# Microstructure Evolution and Fracture Mechanism of 55NiCrMoV7 Hot-Working Die Steel during High-Temperature Tensile

Yasha Yuan [1,2,3], Wenyan Wang [1,*], Ruxing Shi [2,3], Yudong Zhang [2,3] and Jingpei Xie [1]

[1] School of Materials Science and Engineering, Henan University of Science and Technology, Luoyang 471023, China; 15138753540@163.com (Y.Y.); xiejp@mail.haust.edu.cn (J.X.)
[2] CITIC Heavy Industries Co., Ltd., Luoyang 471039, China; 15038647370@163.com (R.S.); zlwzyd07121129@163.com (Y.Z.)
[3] Luo Yang CITIC HIC Casting and Forging Co., Ltd., Luoyang 471039, China
[*] Correspondence: wangwy1963@163.com

**Abstract:** In this paper, through high-temperature tensile tests of 55NiCrMoV7 steel, high-temperature fracture behavior, microstructure evolution, and carbide distribution characteristics of both the thermal–mechanical coupling zone (fracture zone) and thermal stress zone (clamping zone) at different temperatures were studied. Intrinsic relationships between high-temperature fractures and carbide types, distribution and size were revealed, and evolution mechanisms of microstructure near cracks in 55NiCrMoV7 hot-working die steel during high-temperature deformation was clarified. Samples were stretched at different temperatures from 25 °C to 700 °C, and microscopic examinations were carried out using SEM and TEM. The results showed the following. With the increase in temperature, tensile strength and yield strength decreased, elongation and reduction of area increased, and fracture mode changed from brittle fracture to ductile fracture by transition temperature at about 400 °C. During high-temperature deformation, the grain dislocation density decreased and the tempered martensite decomposed, recovered, recrystallized, and then grain grew. M7C3 and M23C6 carbides precipitated and grew along the grain boundary, and a small amount of fine granular MC carbides was dispersed in the grain. The work done by the external force on the deformation zone would cause the temperature of it to be higher than the tensile temperature, which provides thermodynamic conditions for the redissolution of small carbides near the fracture zone and the grain growth of large carbides, resulting in a decrease in small carbides and increase in large carbides in thermal–mechanical coupling zones.

**Keywords:** 55iCrMoV7 steel; hot-working die steel; microstructure evolution; high-temperature fracture mechanism; carbides

## 1. Introduction

Hot-working die steel is used to form high-temperature solid or liquid metal [1,2]. Since the die cavity is in direct contact with the high-temperature metal, and also huge mechanical stress, strong friction caused by the flow deformation of hot metal and thermal stress caused by the alternating action between hot metal and cooling medium, hot-working die steel fails easily during high-temperature service [3–7]. A kind of hot-working die steel with high strength and wear resistance, 55NiCrMoV7 has good impact resistance and tempering stability and is widely used in dies for forgings, such as in aviation, the military, and automobiles [8–11]. As the die industry develops in the direction of large-scale, complex, precise, high-efficiency, and fast-paced applications, its service environment is becoming more and more harsh, which requires higher requirements for die materials [12–15]. In a service environment of 600 °C, due to insufficient thermal strength, steel undergoes high-temperature wear, collapse, sag, and other failure phenomena, which seriously affect

its service life [16]. Many scholars have improved the mechanical properties of die steel by microalloying, adjusting the heat-treatment process, laser cladding, etc. Zhang et al. [17] found that adjusting the content of Mo and V in 5CrNiMo and adding a small amount of Nb could increase the MC content in the steel, thereby greatly improving the thermal stability of the material. Babu et al. [18] adopted cryogenic heat treatment [19] to significantly improve the thermal stability and fatigue properties of steel. Telasang et al. [20–23] believed that laser cladding could greatly improve the hardness, friction, and wear properties of die steel. However, the high-temperature fracture mechanism of 55NiCrMoV7 material has not been studied, so the problem of insufficient thermal strength cannot be solved from the source.

As we all know, mechanical properties are not only related to the original microstructure before stretching but also closely related to the microstructure at fracture, second phase type, size, etc. There have been many studies on the mechanical properties and fracture mechanism of die steel at room temperature, but few on high-temperature fracture mechanisms and the evolution of carbide and microstructure during high-temperature deformation of 55NiCrMoV7 hot-working die steel. In this paper, the mechanical properties of 55NiCrMoV7 steel at different tensile temperatures were studied, and the microstructure near the fracture and in the thermal stress zone were systematically analyzed so as to better understand the microstructure evolution during high-temperature service and fracture mechanisms during high-temperature deformation. The results have theoretical guiding significance for the development of high-strength, large-scale, high-end hot-working die steel under harsh service environments.

## 2. Experimental Materials and Procedure

### 2.1. Materials and Heat Treatment

The type of steel material used in the test was 55NiCrMoV7, with chemical components shown in Table 1. The material preparation process was as follows: smelting 50 kg steel ingot in vacuum induction furnace, followed by annealing–cutting water riser and forging-stress relief annealing.

Figure 1b shows the microstructure of 55NiCrMoV7 steel after heat treatment according to the process in Figure 1a, which was composed of tempered martensite with high-density dislocations, a small amount of recrystallized ferrite subgrain and three kinds of carbides: 30 nm MC, 40–60 nm $M_7C_3$, and $M_{23}C_6$.

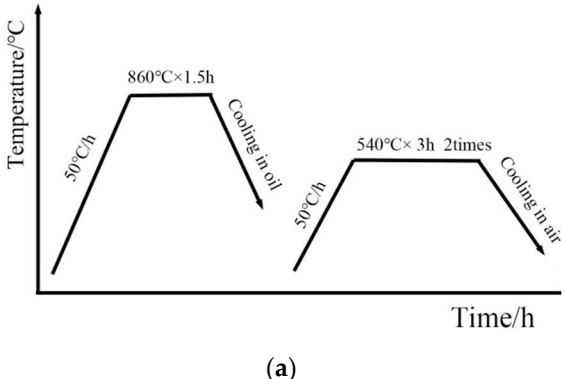

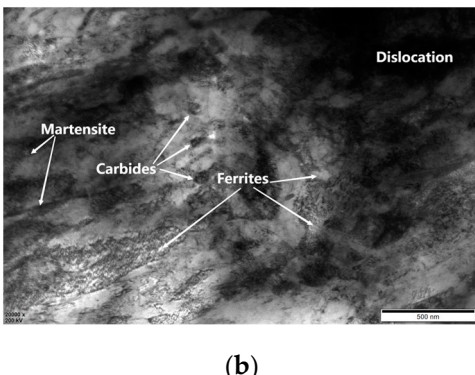

(a)          (b)

**Figure 1.** (**a**) Temperature–time curve of quenching and tempering process; (**b**) microstructure of 55NiCrMoV7 steel after quenching and tempering.

**Table 1.** Chemical composition of the tested 55NiCrMoV7 steel (weight percentage).

| C | Si | Mn | S | P | Cr | Ni | Mo | V |
|---|---|---|---|---|---|---|---|---|
| 0.56 | 0.20 | 0.81 | 0.0010 | 0.0065 | 1.15 | 1.78 | 0.52 | 0.10 |

## 2.2. Mechanical Properties

Tempered tensile samples with dimensions and demonstration shown in Figure 2 were prepared under room temperature and high temperature according to GB/T4338. Room-temperature and high-temperature tensile tests were carried out with a DDL300 high-temperature electronic universal testing machine (Sinter Co., Ltd., Changchun, China) under tensile temperatures of 25 °C, 100 °C, 200 °C, 300 °C, 400 °C, 500 °C, 600 °C and 700 °C. Tensile testing used a high-temperature resistance furnace (WHEF Co., Ltd., Wuhan, CHINA)with temperature difference of ±3 °C to heat the sample. After reaching the set temperature, samples were kept for 20 min to ensure that the temperature of the surface and the core were uniform. After this heat preservation, tensile tests were carried out at 1 mm/min, and samples were air-cooled after breaking. Three samples were tested at each temperature and the average value was taken. After the test, the tensile strength and yield strength were obtained. Elongation and reduction in area were calculated from the difference between effective gauge length and diameter of the tensile samples before and after the test. The A area of the sample was the thermal stress zone and B area the thermal–mechanical coupling zone, as shown in Figure 2.

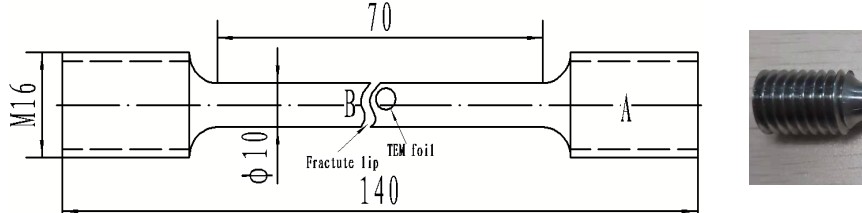 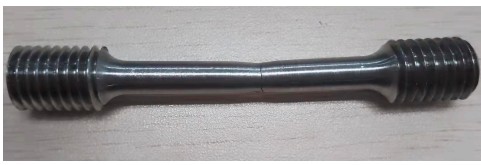

**Figure 2.** Dimensions and demonstration of tensile samples.

## 2.3. Characterization of Microstructure

In order to observe the structural changes near the fracture, a small metallographic sample near the fracture perpendicular to the tensile direction (B area in Figure 2) was taken out and prepared after mechanical grinding, polishing, and 4% nitric acid alcohol corrosion. The tensile fracture morphology and microstructure of the samples were observed by SM-5610LV SEM. A JEM-2100 HRTEM TEM (Carl Zeiss AG, Oberkochen, Germany) was used to analyze microstructures of different type, size and distribution of the second phase, and the size of carbide was characterized and recorded.

## 3. Result and Discussion

### 3.1. High-Temperature Mechanical Properties of 55NiCrMoV7 Steel

Figure 3 shows the high-temperature performance curves of 55NiCrMoV7 steel at different tensile temperatures after oil-cooling for 860 °C × 1.5 h and secondary tempering for 540 °C × 3 h. It can be seen from the figure that the tensile strength of 55NiCrMoV7 steel was the highest at room temperature, which was 1490 MPa. With the increase in temperature, the tensile strength decreased. It decreased slightly when the temperature was lower than 400 °C and decreased rapidly when the temperature exceeded 400 °C. The tensile strength was only 107 MPa at 700 °C. The trend of high-temperature tensile yield strength of 55NiCrMoV7 steel was similar to that of tensile strength. The difference between tensile strength and yield strength was only 17 MPa at 700 °C. On the contrary, the elongation and reduction in area increased with the increase in temperature. When the temperature exceeded 400 °C, the rate of increase increased significantly and the elongation and reduction of area increased from 11% and 25% at room temperature to 47% and 90% at 700 °C, respectively.

### 3.2. High-Temperature Tensile Fracture Morphology

Figure 4 shows the fracture morphologies of 55NiCrMoV7 steel at different tensile temperatures. It can be seen from Figure 4a that the tensile fracture at room temperature had

no obvious necking phenomenon, and the fracture was relatively flat and perpendicular to the tensile axis. There were secondary cracks in the fracture, which started at the center of the sample. Under tensile stress, the cracks rapidly expanded from the center of the sample and broke instantaneously. The fracture was mainly composed of a large number of cleavage planes and a few shear-like dimples [24]. The room temperature tensile fracture of 55NiCrMoV7 steel was brittle.

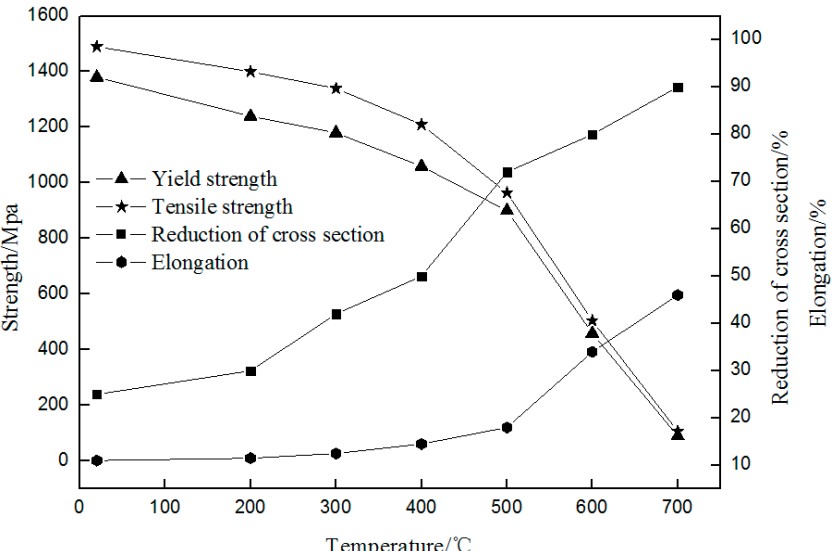

**Figure 3.** Curve of high-temperature mechanical properties of 55NiCrMoV7 steel.

The high-temperature tensile fracture of 55NiCrMoV7 steel exhibited obvious ductile fracture characteristics, such as necking and cup–cone. When the tensile temperature was 400 °C, the crack initiation zone, crack propagation zone and shear lip zone could be clearly observed, among which the crack propagation zone occupied the largest fracture area and a large number of equiaxed dimples with size of 0.8–1.2 μm could be seen in the fracture. When the stretching temperature increased from 400 °C to 500 °C, a few 1.8–2.6 μm plastic pores appeared in the fracture, while the rest of the fracture characteristics did not change much. The fracture area was reduced, which was consistent with the higher reduction in area at 500 °C in Figure 3. When the tensile temperature rose to 600 °C, tensile deformation caused the cross section to be at a 45° angle to the tensile direction, resulting in shear stress. With the increase in stress, crack propagation, penetration and failure occurred, and the plastic deformation characteristics near the fracture were more obvious; at the same time, a large number of scaly slippage bands appeared on the surface of the specimen near the fracture. Compared with low temperature, the dimple depth in the fracture became deeper and dimple size increased obviously and showed obvious ductile fracture which was consistent with the low strength and high toughness results shown in Figure 3 at this temperature.

From fracture analysis at different tensile temperatures, it could be concluded that as the tensile temperature increased, the tensile fracture changed from brittle to ductile, dimple numbers gradually decreased and became larger from 0.8–1.2 μm at 400 °C to 2.8–4.2 μm at 600 °C.

Table 2 shows the statistical results of the area of fracture crack propagation zone and crack initiation zone at different tensile temperatures. It can be seen that the area of the crack propagation zone in the fracture at different temperatures was larger than the area of the crack initiation zone, and there was a positive correlation between the two areas and the tensile strength of steel. The larger two areas effectively hindered the formation and propagation of cracks and improve crack formation energy. Under tensile stress, the crack reached the instantaneous breaking area and rapidly expanded and broke along the direction of the maximum shear force. Therefore, a large crack initiation area and crack propagation area was

beneficial to improve the high-temperature tensile strength of steel, which was consistent with the mechanical properties at high temperature shown in Figure 3.

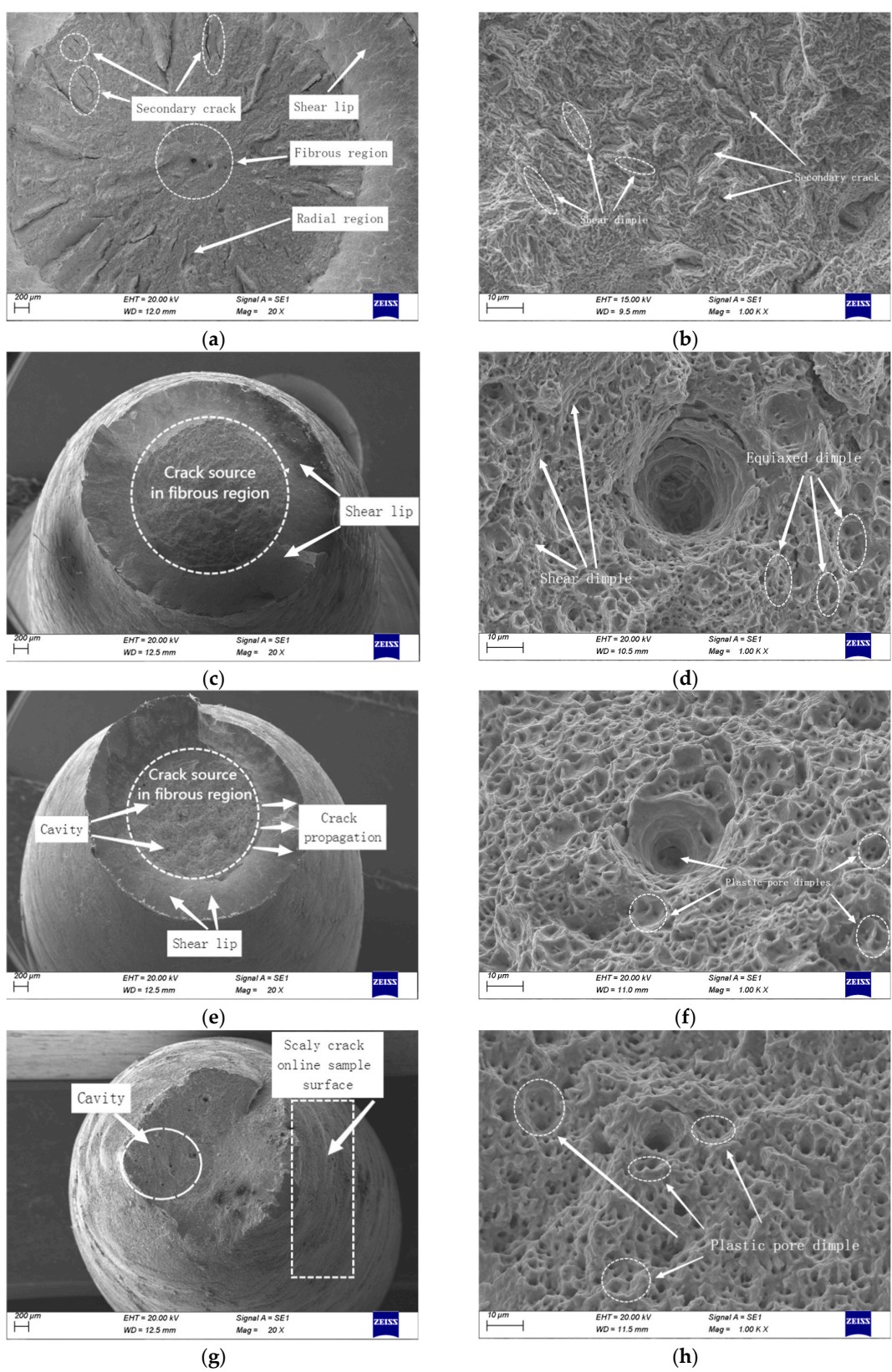

**Figure 4.** Tensile fracture morphology of 55NiCrMoV7 steel at different temperatures. (**a**,**b**) Room temperature; (**c**,**d**) 400 °C; (**e**,**f**) 500 °C; (**g**,**h**) 600 °C.

**Table 2.** Statistical results of the area of fracture crack propagation zone and crack initiation zone.

| Tensile Temperature/°C | The Area of Fracture Crack Propagation Zone/mm² | The Area of Crack Initiation Zone/mm² | The Total Area/mm² |
|---|---|---|---|
| Room temperature | 16.05 | 2.10 | 18.15 |
| 400 °C | 6.94 | 0.92 | 7.86 |
| 500 °C | 5.43 | 0.69 | 6.12 |
| 600 °C | 3.08 | 0.31 | 3.39 |

### 3.3. Effect of Tensile Temperature on Microstructure of 55NiCrMoV7 Steel

Figure 5 shows the TEM images of 55NiCrMoV7 steel near the fracture at different tensile temperatures. It can be seen from the figure that the microstructure was tempered lath martensite with a width of about 160 nm and a length of 1600 nm after stretching at room temperature. There were a large number of high-density dislocations distributed in the martensite, and dislocation slip could only happen in a limited distance, which was the main reason for the high strength and low toughness of 55NiCrMoV7 steel at room temperature.

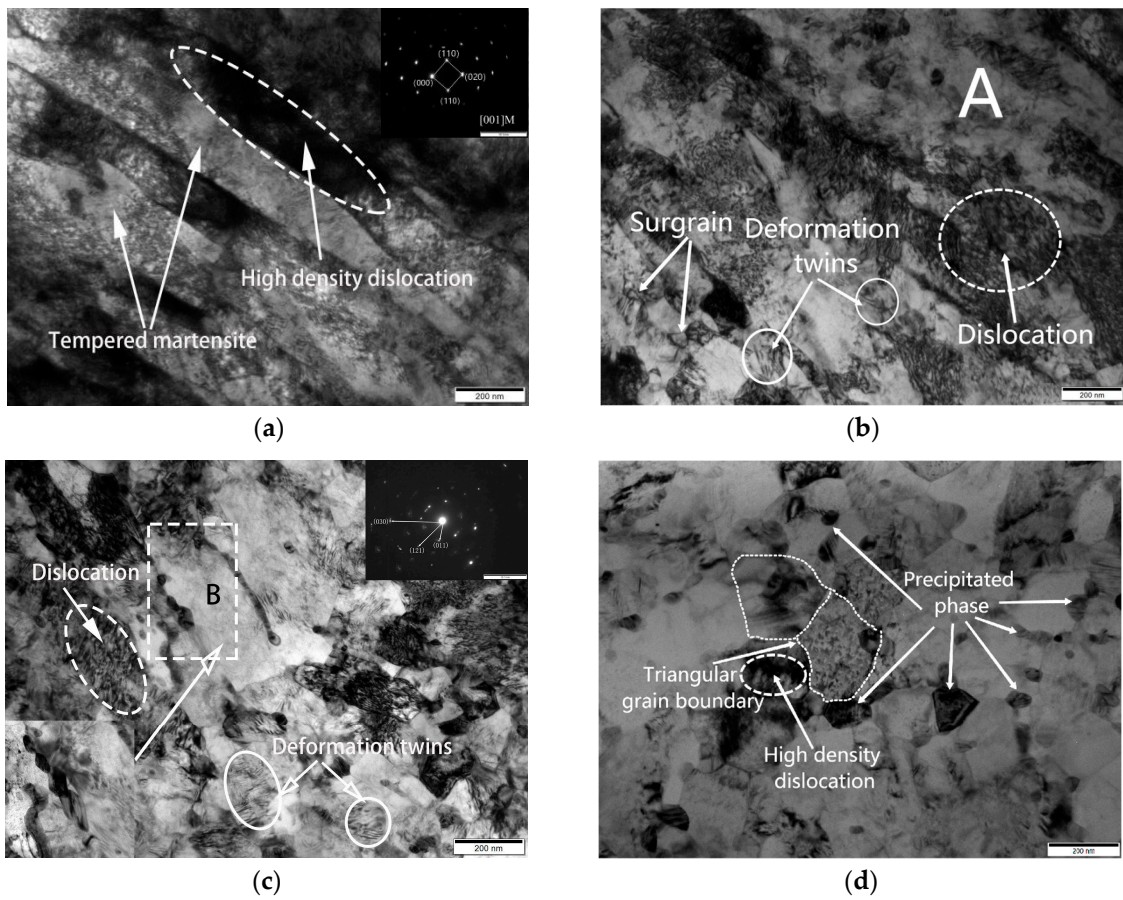

**Figure 5.** TEM images of 55NiCrMoV steel near the fracture at different tensile temperatures. (**a**) Room temperature; (**b**) 400 °C; (**c**) 500 °C; (**d**) 600 °C.

At a stretching temperature of 400 °C, the tempered lath martensitic still existed in some areas (shown in Area A in Figure 5b). A small amount of deformed twin crystals could be observed in the tempered martensite beam, which had a significant effect on improving the plasticity and toughness of the material. Compared with room temperature stretching, the dislocation density and the degree of dislocation entanglement were significantly reduced, and 20–50 nm subgrains formed by recovery and recrystallization were found in some regions.

When stretching temperature was further increased to 500 °C, dislocation density was also further reduced, and the recovery–recrystallization phenomenon was more obvious, all resulting in a significant decrease in the strength of the material. A large number of twin crystals appeared in the grains and showed obvious recrystallization. According to the diffraction spot (shown in the upper right corner of Figure 5c), it could be seen that a large number of $\alpha$-phase polycrystalline diffraction rings were formed at this temperature and dislocation density was significantly reduced. Only a small part of the remaining martensite showed wave-like discontinuous tempering deformation characteristics, strong evidence of strength reduction under high-temperature tensile. A large number of equiaxed grains in the microstructure were produced by grain breakage and recrystallization under high-temperature tensile stress, and the second phase was preferentially precipitated at the martensite lath boundary (shown in lower left corner of Figure 5c and Area B), reducing the continuity between grains. Under tensile stress, the stress concentration around the second phase weakened the grain boundary strengthening effect, accelerated the crack propagation, and reduced the strength of the material.

When the stretching temperature reached 600 °C, the recovery and recrystallization of tempered martensite were basically completed, the subgrains formed by recrystallization grew significantly, dislocations basically disappeared, and only a small number of high-density dislocations existed at local trigeminal grain boundaries. The number and size of carbides precipitated along the grain boundary increased significantly.

*3.4. Precipitate Phase Analysis*

Figure 6 shows TEM images and selected electron diffraction pattern of $M_{23}C_6$, $M_7C_3$ and MC carbides in 55NiCrMoV7 steel when stretched at 500 °C. It can be seen from Figure 6a that carbides of 30–70 nm width and 100 nm length elongated along the grain boundaries were precipitated at the grain boundaries. Through the calibration of the electron diffraction pattern, they were $M_{23}C_6$ carbide (Figure 6b). It can also be seen from Figure 6c that there were sharp-angle precipitates with a width of about 100 nm and length of 200 nm at the intersection of the martensitic lath boundary. Through the calibration of the electron diffraction pattern, they were $M_7C_3$ carbide mainly containing Cr (Figure 6d). Compared with $M_7C_3$ (40~60 nm) in unstretched 55NiCrMoV7 steel, the carbides grew significantly under tensile stress at high temperature. This was mainly due to the fact that under the tensile stretch at 500 °C, the tempered martensite underwent a decomposition–recovery–recrystallization process, and the dislocations were pushed to the surrounding grain boundaries to form dislocation walls, which provided dynamic conditions for precipitation and growth of carbides at grain boundaries. The work done by the external force on the deformation zone during high-temperature stretching caused the temperature of it to be higher than tensile temperature, which provided thermodynamic conditions for the redissolution of small carbides near the fracture zone and the grain growth of large carbides [25,26]. Due to the low thermal activation energies of $M_7C_3$ and $M_{23}C_6$ carbides [27], Cr-containing carbides grew significantly during high-temperature stretching.

It can be seen from Figure 6e that there were small and dispersed carbides in the grains, with a size of about 30 nm. It can be seen from the electron diffraction pattern that they were MC carbides (Figure 6f). They were similar to the size of MC in unstretched 55NiCrMoV7 steel, which was mainly due to the larger growth activation energy and higher thermal stability of MC carbides [28,29].

*3.5. Microstructure of Thermal Stress Zone and Thermal–Mechanical Coupling Zone of High-Temperature Stretching Samples*

Figure 7 shows the microstructure of the thermal stress zone and thermal–mechanical coupling zone at different stretching temperatures. When tensile temperature was 400 °C, the microstructure of the thermal stress zone was composed of tempered martensite and a small amount of nanosized carbides. Under tensile stress, the microstructure of the thermal–

mechanical coupling zone showed obvious deformation and directionality. When tensile temperature reached 600 °C, the large carbides increased significantly and grew. Martensite decomposed, the microstructure showed obvious recrystallization, and the ferrite content increased slightly. Compared with the thermal stress zone, plastic deformation in the thermal coupling zone was more obvious, small carbides reduced, and large carbides increased, Under the action of thermal–mechanical coupling, the stress accumulated easily at the sharp corners of carbide, and the sharp corners of large carbide gradually dissolved under the action of high stress, showing that the carbide morphology was relatively circular in the thermal–mechanical region.

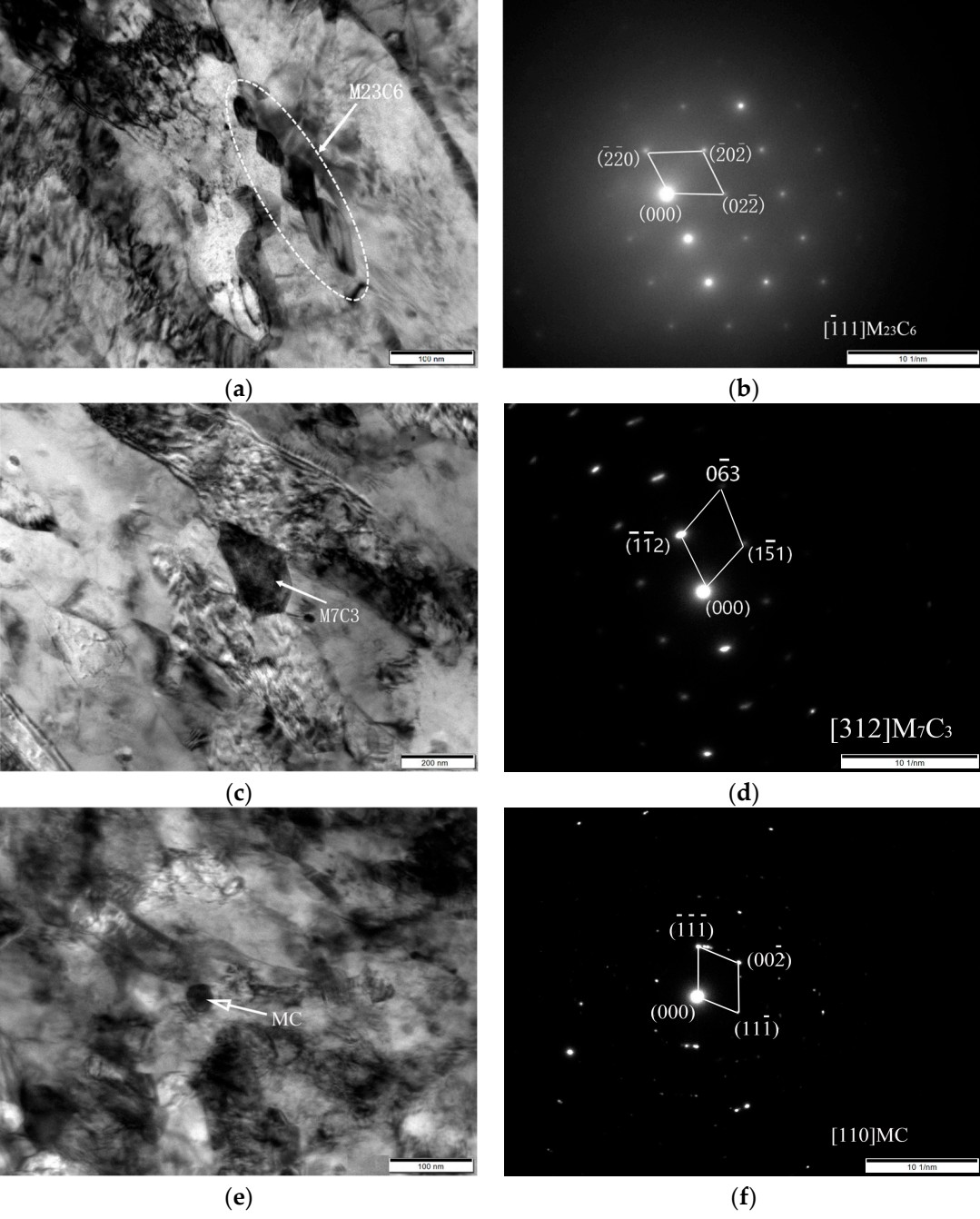

**Figure 6.** TEM images and selected electron diffraction pattern of precipitated phase of 55NiCrMoV steel at 500 °C stretching. (**a**) Bright-field image of $M_{23}C_6$; (**b**) selected electron diffraction pattern of $M_{23}C_6$; (**c**) bright-field image of $M_7C_3$; (**d**) selected electron diffraction pattern of $M_7C_3$; (**e**) bright-field image of MC; (**f**) selected electron diffraction pattern of MC.

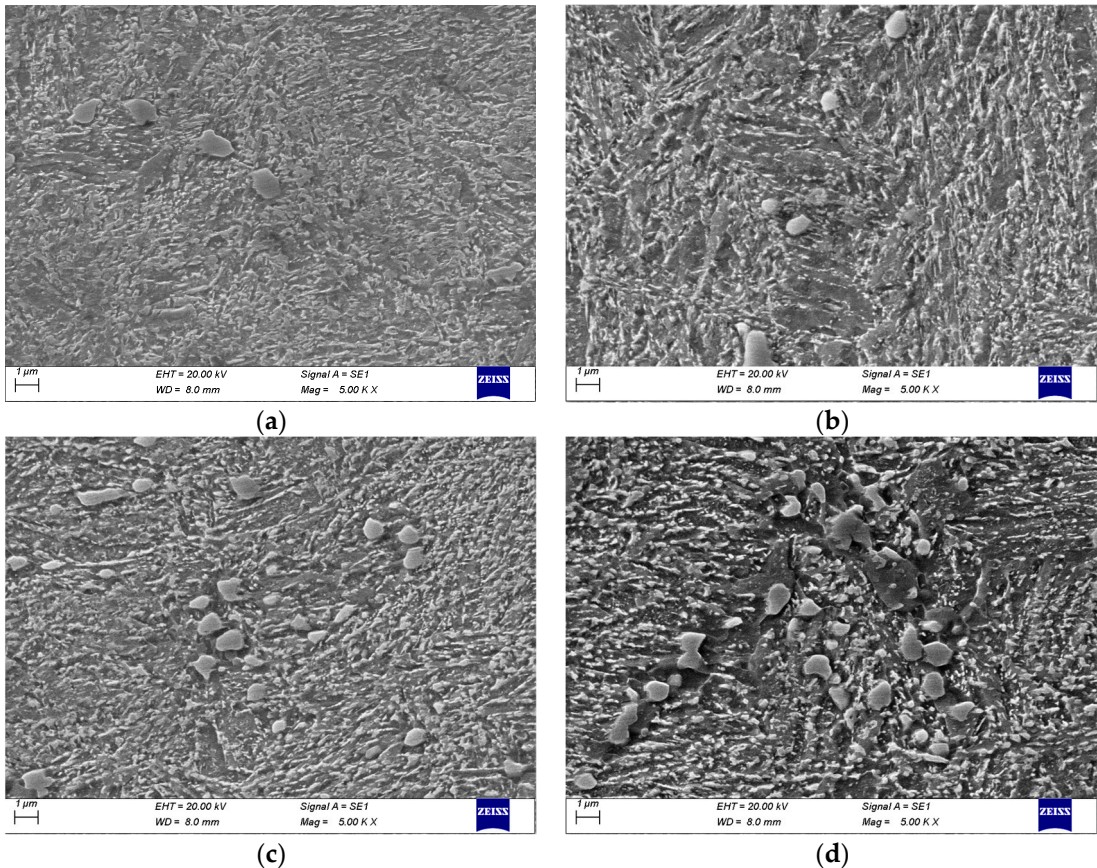

**Figure 7.** Microstructure of thermal stress zone and thermal–mechanical coupling zone at different stretching temperatures. (**a**) Thermal stress zone at 400 °C; (**b**) thermal–mechanical coupling zone at 400 °C; (**c**) thermal stress zone at 600 °C; (**d**) thermal–mechanical coupling zone at 600 °C.

Figure 8 shows the number and size of carbides in the thermal stress zone and thermal–mechanical coupling zone under different conditions. It can be seen from the figure that the number of small carbides in the thermal–mechanical coupling zone was fewer than in the thermal stress zone. The thermal–mechanical coupling effect led to a reduction in carbides. Compared with different temperatures, there were more small carbides and fewer large carbides in the thermal stress zone and thermal–mechanical coupling zone at 400 °C than 600 °C. Comparing the thermal stress and thermal–mechanical coupling zone, it could be seen that small carbides below $0.2 \times 10^2$ nm were redissolved under thermal–mechanical coupling, the number of $(0.2–1.0) \times 10^2$ nm carbides was significantly reduced, and the number of $(1.0–1.5) \times 10^2$ nm carbides remained unchanged. The sharp corners of large carbides decomposed and gradually became rounded, and the percentage of carbides above $2.0 \times 10^2$ nm increased significantly.

Under the thermal–mechanical coupling effect, the dislocation proliferation caused by tensile deformation provided a channel for the diffusion of carbon atoms and alloying elements. At 600 °C, the strengthening effect of solid solution and dislocation weakened, which led to an increase of plasticity of steel, tensile deformation, and the tensile force formed an angle of 45°, resulting in shear stress. Small carbides redissolved and large carbides aggregated to form clusters. Due to the soft matrix and the matrix between clusters having no carbide pinning dislocations, large and deep plastic pores were formed under tensile stress. Therefore, the steel had high elongation and reduction in area at 600 °C.

When the temperature increased from 400 °C to 600 °C, the solid-solution strengthening and dislocation strengthening of the steel weakened, resulting in a decrease in strength and increase in plasticity of the steel. The bonding energy between metal atoms decreased the discontinuous stress distribution between carbide and matrix and led to uneven defor-

mation of the steel, and the fracture morphology showed dimples with different depths. With the increase in tensile temperature, the dislocation density decreased obviously, the precipitated carbides grew, and under tensile stress, cracks began to initiate near the carbides. Without the strengthening effect of high-density dislocations, the carbides were insufficient to withstand large stresses and breaks, which was consistent with the results of low-strength high plastic at 600 °C (Figure 3).

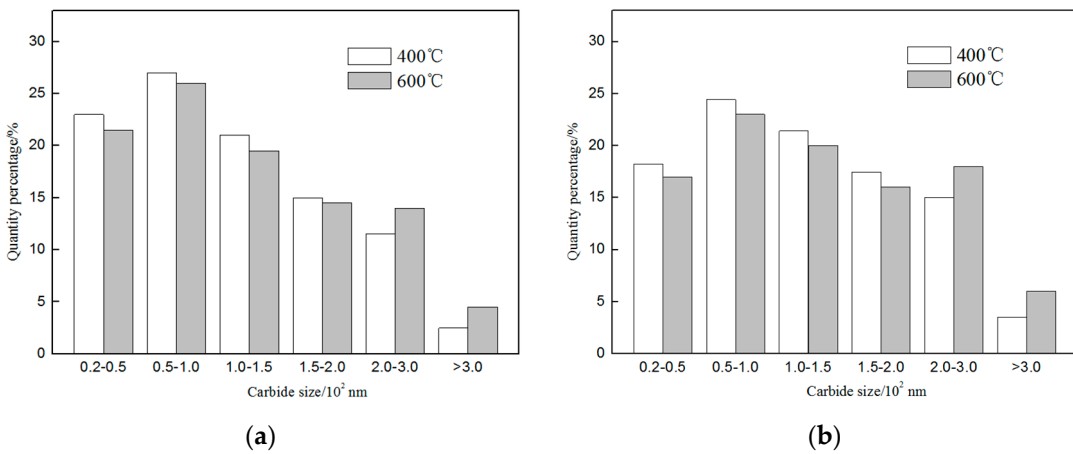

**Figure 8.** Carbide size distribution of 55NiCrMoV7 steel at different tensile temperatures. (**a**) Thermal stress zone; (**b**) thermal–mechanical coupling zone.

### 3.6. Effect of Carbides on Crack Initiation and Propagation

In order to further obtain the relationship between carbides and crack initiation and propagation, the microstructure and potential cracks of the 500 °C tensile samples near the fracture were studied in detail. As shown in Figure 9, there were cracks of different lengths near the fracture and one of the long cracks was analyzed. There were approximately rod-shaped (irregular shape and small sharp corners) and spherical carbides distributed in the middle and end of the cracks, respectively. The composition changes and distributions of Fe, C, Cr, and Mn shown in Figure 9b were obtained by performing line scanning through the rod carbide along the AB line in Figure 9a. The composition change and distribution of Fe, C, Mo and V shown in the upper right corner of Figure 9a were obtained by EDS energy spectrum analysis of the D carbide in Figure 9a. According to the distribution of alloying elements and previous tests, the rod-shaped carbide C was $M_{23}C_6$ or $M_7C_3$ carbide mainly containing Cr and Mn, and the spherical carbide D was MC carbide mainly containing Mo and V. It can be seen from the figure that when the crack started near the $M_{23}C_6$ carbide, the crack continued to expand along the C carbide interface, and when the crack extended near the D carbide, the crack stopped expanding, indicating that the approximate spherical MC carbide had a blocking effect on the crack propagation.

In Cr-Mo-V materials, $M_{23}C_6$ and $M_7C_3$ carbides maintained an incoherent relationship with the matrix with low thermal activation energy. Under mechanical and thermal stress, they precipitated along the grain boundary and were easy to grow. Due to irregular shape, stress concentration occurred easily at sharp corners and crack sources generated easily. MC carbides were finely dispersed in the matrix, and their shape was nearly spherical, which was not conducive to stress concentration. MC carbides maintained a coherent relationship with the matrix, had high thermal activation energy, were not easy to grow, could pin dislocations, hindered dislocation movement, reduced stress concentration generated by dislocations at large particle carbides, and effectively prevented crack initiation and propagation.

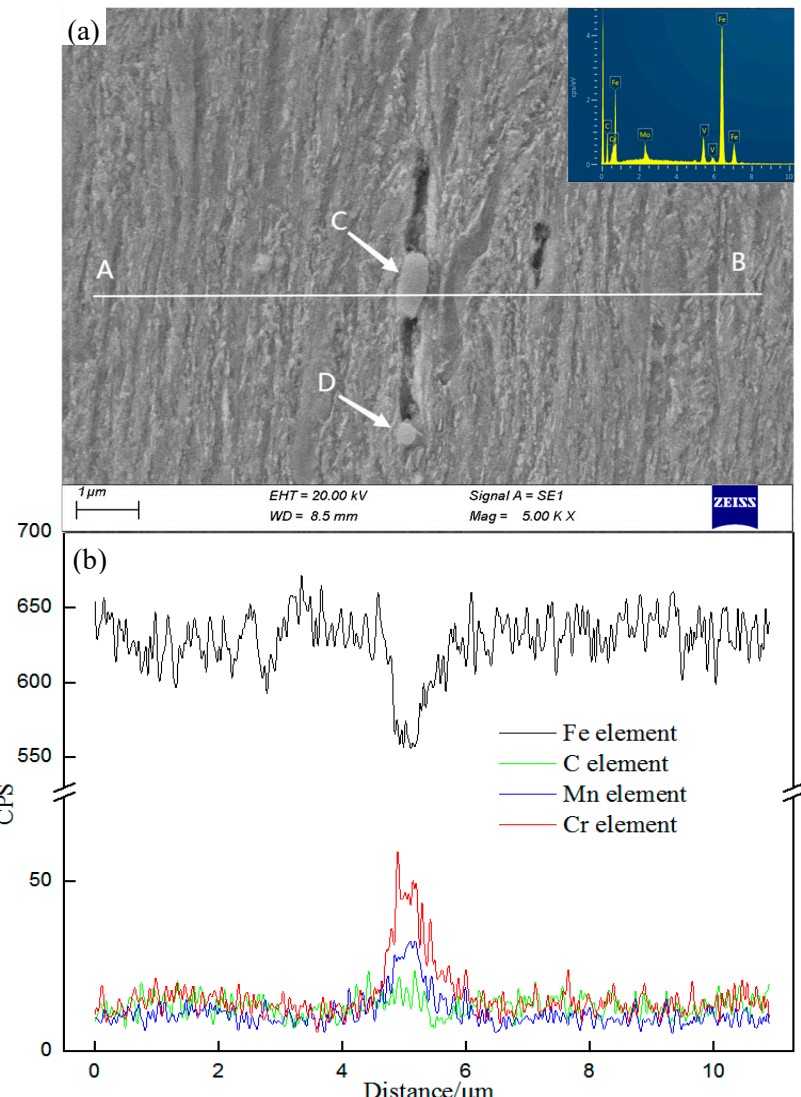

**Figure 9.** Line scanning of Fe, C, Mn and Cr elements near the crack in 55NiCrMoV7 steel. (**a**) SEM image of crack propagation; (**b**) line scanning and element EDS results.

## 4. Conclusions

(1) With tensile temperature increased from room temperature to 700 °C, the tensile strength of 55NiCrMoV7 steel decreased from 1490 MPa to 107 MPa, and the elongation and reduction of area increased from 11% and 25% to 47% and 90%, respectively. When stretching temperature was higher than 400 °C, the decrease rate of strength and the increase rate of plasticity increased significantly.

(2) In this experiment, the tensile fracture of 55NiCrMoV7 steel changed from brittle fracture to ductile fracture by transition temperature of 400 °C. The dimple size gradually increased from 0.8–1.2 μm at 400 °C to 2.8–4.2 μm at 600 °C with the increase in tensile temperature.

(3) The microstructure under room temperature stretching was mainly lath martensite with many high-density dislocations distributed inside. During high-temperature deformation, the microstructure showed obvious recovery and recrystallization, and the dislocation density decreased. The $M_7C_3$ and $M_{23}C_6$ carbides that were incoherent to the matrix precipitated and grew along the grain boundaries, while other MC carbides that were coherent to the matrix were dispersed in the grains.

(4) Under high-temperature tensile stress, stress concentration was easily generated around the $M_7C_3$ and $M_{23}C_6$ carbides precipitated along the grain boundary, which

weakened the strengthening effect of the grain boundary and accelerated the crack propagation, while the granular MC carbides dispersed in the grain could prevent cracks. Therefore, reducing $M_7C_3$ and $M_{23}C_6$ carbide content and increasing other MC carbide content were beneficial in improving high-temperature performance of hot-working die steel.

(5) During high-temperature stretching, the work done by the external force on the deformation zone would cause the temperature of it to be higher than tensile temperature, which provided thermodynamic conditions for the redissolution of small carbides near the fracture zone and the grain growth of large carbides, resulting in the decrease in small carbides and increase in large carbides in the thermal–mechanical coupling zone.

**Author Contributions:** Conceptualization, Y.Y., R.S. and J.X.; Formal analysis, Y.Y. and Y.Z.; writing—original draft, Y.Y. and Y.Z.; writing—review and editing, W.W.; supervision, W.W. and J.X.; project administration, Y.Z.; funding acquisition, R.S. All authors have read and agreed to the published version of the manuscript.

**Funding:** The authors acknowledge with gratitude funding received from the National Key R&D Program of China (grant 2020YFB2008400) and Major Science and Technology projects (221100230200).

**Informed Consent Statement:** Informed consent was obtained from all subjects involved in the study.

**Data Availability Statement:** Because data is unavailable due to privacy, Where no new data were created.

**Conflicts of Interest:** The authors (Yasha Yuan, Wenyan Wang, Ruxing Shi, Yudong Zhang, Jingpei Xie) declare that we have no financial and personal relationships with other people or organizations that could inappropriately have influenced our work, there is no professional or other personal interest of any nature or kind in any product, service and/or company that could be construed as influencing the position presented in or review of this manuscript.

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
