# Peer review of "Microstructure Evolution and Fracture Mechanism of 55NiCrMoV7 Hot-Working Die Steel during High-Temperature Tensile"

_metals, doi:10.3390/met13061056_

Round 1
Reviewer 1 Report
In this manuscript, the authors investigated the mechanical and fracture behaviour of 55NiCrMoV7 steel subjected to deformation at different temperatures were investigated. The manuscript lacks the depth in discussion. Below are some issues with the writing.
1) In the Introduction section, the motivation of using 55NiCrMoV17 steel for this study was not stated. What are the issues with this steel that there is a need to study it’s microstructure and fracture behaviour?
2) There is a need to perform a gap analysis based on the literatures to highlight the novelty of this work.
3) Lines 67 to 70, how did you obtain this information? How did you know that the steel composed of “high density dislocation”?
4) Please state how dd you obtained the properties given in Table 1?
5) Line 74, Please elaborate what is GB/T4338.
6) Line 101, “ … after oil cooling for 860 ℃×1.5h and secondary tempering for 540 ℃×3h…” This is confusing as this process was not stated in the Experimental section. The sample preparation and all experimental procedures should be clearly highlighted in the Experimental section.
7) Line 141 to 144, “From fracture analysis at different tensile temperatures, it could be concluded that as the tensile temperature increased, the tensile fracture changed from brittle fracture to ductile fracture, dimple numbers gradually decreased, dimple sizes became larger …” So why is this the case? What is the role of the temperature in governing the mechanical behaviour of the steel?
8) Line 168 to 170, “There were a large number of high-density dislocations distributed in the martensite, and dislocation slip could only happen in a limited distance, which was the main reason for the high strength and low toughness of 55NiCrMoV7 steel at room temperature..” This is speculative in nature and is not obvious from the TEM pictures.
9) Line 173, “A small amount of deformed twin-crystals could be observed in the tempered martensite beam …” This is also speculation.
10) Line 180 – 182, speculation statements … what evidence have you to indicate that the supersaturated carbon was re-distributed starting from 500°C onwards and not below this temperature?
11) Line 195 – 199, this is also speculation
12) Line 204, please explain the reason for the preferential carbide precipitation at the grain boundaries and not at other section of the grain?
13) Line 211 to 213, another speculation statement.
14) Line 240 to 243, another speculation on the carbides behaviour. How do you know that there is a reduction in the alloying elements in the grain?
15) Line 265 to 267, another speculation on the carbon and alloying elements without providing any proof.
16) Line 271, “Due to the soft matrix, large and deep plastic pores were formed under tensile stress.” How is this so? Please elaborate the formation of plastic pores ….
Reviewer 2 Report
This paper mainly addressed the high temperature tensile effect on the microstructure and fracture mechanisms. The strength of this paper is that the approach to investigate the fracture mechanisms give good novelty compared with other published material. Room for improvement:
1. Please check for superscript for the authors affiliations
2. The microstructure after heat treatment for Fig 1 is missing. Labels within figure 1 and 2 are not clear
3. Its better to introduce what are MC, M7C3, M23C6.
4. "Imgae Pro" is the name correct?
5. MPa instead of Mpa
6. Fig. 4c: typo on the label: "fibous"
7. Page 7, line 213-217: the sentences related are just speculating the effect of external force on the zones. Some citations from literature should be included to strengthen the argument.
8. Is temperature 500-600C are sufficient to allow the small carbides to re-dissolved? How to justify this?
9. Please check the references bibliography. Some of the paper listed are still containing "et al".
Reviewer 3 Report
Dear Authors. Your paper on high-temperature tensile tests on 55NiCrMoV7 steel to investigate the high-temperature fracture properties, microstructure evolution, and carbide distribution characteristics of both the thermomechanical coupling zone (fracture zone) and the thermal stress zone (clamp zone) at different temperatures was a good read. I have no complaints, just a few suggestions for improvement.
Your manuscript deals with the high-temperature fracture behavior of the steel 55NiCrMoV7. As a hot work tool steel, the material properties with respect to thermomechanical failure at different temperatures are of particular importance.
High-temperature tensile tests are used to investigate the microstructural evolution and carbide distribution properties resulting from the thermomechanical stress in the fracture and stress zones.
The microscopic investigations were performed classically with a scanning and transmission electron microscope. The relationship between temperature and tensile strength, yield point and elongation can be graphically illustrated.
The investigations carried out by you extend our knowledge of the fracture behavior and the underlying mechanisms of the steel 55NiCrMoV7. The relevant information is provided by a sufficient number of references.
The findings reported in the conclusion reflect the experimental results described in the previous text. Although I am not directly involved in the subject of hot work steels, I can assume that this work will deepen our knowledge and stimulate further, more detailed investigations.
Don't you abbreviate megapascals to MPa?
The text within the figures in Figure 4 and Figure 8 is difficult to read.
Should line 208 read Fig 6a,b and 6c,d?
Round 2
Reviewer 1 Report
The authors have satisfactorily addressed the reviewer's comments.